# The Effect of Ni-Modified LSFCO Promoting Layer on the Gas Produced through Co-Electrolysis of $CO_2$ and $H_2O$ at Intermediate Temperatures

Massimiliano Lo Faro *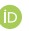, Sabrina Campagna Zignani, Vincenzo Antonucci and Antonino Salvatore Aricò

Institute of Advanced Energy Technologies "Nicola Giordano" (ITAE) of the Italian National Research Council (CNR), Via Salita S. Lucia sopra Contesse 5, 98126 Messina, Italy; zignani@itae.cnr.it (S.C.Z.); vincenzo.antonucci@itae.cnr.it (V.A.); arico@itae.cnr.it (A.S.A.)
* Correspondence: lofaro@itae.cnr.it; Tel.: +39-090-624-243

**Abstract:** The co-electrolysis of $CO_2$ and $H_2O$ at an intermediate temperature is a viable approach for the power-to-gas conversion that deserves further investigation, considering the need for green energy storage. The commercial solid oxide electrolyser is a promising device, but it is still facing issues concerning the high operating temperatures and the improvement of gas value. In this paper we reported the recent findings of a simple approach that we have suggested for solid oxide cells, consisting of the addition of a functional layer coated to the fuel electrode of commercial electrochemical cells. This approach simplifies the transition to the next generation of cells manufactured with the most promising materials currently developed, and improves the gas value in the outlet stream of the cell. Here, the material in use as a coating layer consists of a Ni-modified $La_{0.6}Sr_{0.4}Fe_{0.8}Co_{0.2}O_3$, which was developed and demonstrated as a promising fuel electrode for solid oxide fuel cells. The results discussed in this paper prove the positive role of Ni-modified perovskite as a coating layer for the cathode, since an improvement of about twofold was obtained as regards the quality of gas produced.

**Keywords:** valorisation of $CO_2$; solid oxide electrochemical cells; green methane; energy storage; power-to-gas





## 1. Introduction

The most advanced countries are currently facing environmental threats rising from climate-changing emissions. Their actions are addressed towards a rational use of energy, adopting a circular economy, and acting on environmental restoration [1–3]. In this context, the concept of "what goes around comes around," sounds more than ever as an action point for societies, policy makers, and scientists [4]. As regards the research, one possible action is the capture, reuse and valorisation of the $CO_2$ produced by many industrial sectors, and the following production of fuels [5]. If this action is coupled with the need to store renewable electricity, the derived fuels belong to a sustainable vision and are fully flagged as "green" [6]. A possible way to achieve this target is the use of solid oxide electrochemical cells (SOECs)-based technology, which has been used for the conversion of $H_2O$ and $CO_2$ into syngas at high temperatures [7–9]. Currently, commercial SOECs are manufactured with the same materials and architecture as solid oxide fuel cells (SOFCs), simply because of their robustness [10–14]. However, the use of metallic Ni combined with yttria-stabilised zirconia (YSZ) as a fuel electrode (cathode) has an effect on the quality of the outlet gas. In practise, under conventional operating conditions, the fuel in the outlet gas stream is constituted only of $H_2$ and CO [15–18]. Furthermore, SOECs fed with $CO_2$ and $H_2O$ were operated at high temperatures (above 800 °C) to minimise the effect of carbon deposition on Ni [19,20]. However, the high temperatures promote the agglomeration of Ni [21,22] and suppress the presence of methane, as it is thermodynamically stable

only at lower temperatures [23,24]. The commercial approach used to increase the value of syngas and to decrease the intrinsic risk of CO injection into the gas pipeline consists of the use of a post-chemical processor operating in the range of temperatures between 200 °C and 400 °C [25–27]. This approach is exactly the same as the combination of an SOFC with an external reformer, and implies an increase in the complexity and risks of the overall system [28–31]. An emerging option to improve the chemical reactions occurring on both SOECs and SOFCs cells consists of the addition of a functional layer coated on the fuel electrode. This option is beneficial for the overall system since the dimension and management of the system are simplified. Furthermore, the addition of a coating layer on the fuel electrode does not imply any strong change in the manufacturing chain of cells and stacks. The use of cermets (i.e., a combination between metallic and ceramic phases) is the preferred approach, owing to the physico-chemical compatibility between these materials and the fuel electrode of solid oxide cells (SOCs) [32–34].

Based on the results achieved with SOFCs, the authors have suggested a similar approach for the SOECs to enhance methane production. In 2020, two studies published in the *Journal of Energy Storage* [35] and in the *International Journal of Hydrogen Energy* [36] reported on the findings of Ni–Fe and Cu–Sn alloys combined with CGO as possible functional layers for commercial SOC cells. By comparing gas chromatographic analyses of the outlet gases of these two cells and of a bare cell, it was found that the quality of the outlet gas can be improved by operating at intermediate temperatures and through the simple addition of a functional layer to a commercial SOEC cell.

However, a new class of materials corresponding to "exsolved perovskite" has been recently used as fuel electrodes for SOC cells [37–43]. These materials have a unique morphology based on a substrate with mixed ionic and electronic conductivity (MIEC) supporting encapsulated fine particles that originate from the segregation of metals from the MIEC's bulk [44,45]. As a consequence of this exotic structure, these materials show bi-functional properties (i.e., catalytic and electrocatalytic properties), and these are additional to the well-known properties of perovskite as regards the resilience to organic-based fuels, the redox properties and the resistance to sulphur contaminations [46]. Above all, the Ni-modified $La_{0.6}Sr_{0.4}Fe_{0.8}Co_{0.2}O_3$-based perovskite is one of the most known exsolved perovskites under study. Its electrochemical properties were evaluated in a fuel flexible SOFC [47] and in an all-perovskite-based SOFC [48]. Here, this paper is devoted to the findings concerning the gas quality of a SOEC cell coated with the Ni-modified $La_{0.6}Sr_{0.4}Fe_{0.8}Co_{0.2}O_3$-based perovskite for the co-electrolysis of $H_2O$ and $CO_2$. As such, in this study we report on the preliminary tests conducted by coupling the electrochemical diagnosis of cell and gas outlet analyses. The results are discussed considering our previous achievements with a bare cell investigated under the same conditions [49,50].

## 2. Results and Discussion

The electrochemical tests were carried out for 80 h in a range of temperatures between 525 and 800 °C, and by moving the cell between the open circuit voltage and the operation condition at 150 mA cm$^{-2}$. To achieve a stable condition, we kept the cell in each condition for a minimum of 5 h since the GC analysis of effluent gas required 10 min and the measures were repeated five times. In some case we kept the cell for more time according to the night rests. At the end point of each thermal and current condition, the polarisation curve (current/potential, I/V) and impedance spectroscopy (EIS) at open circuit voltage (OCV) and at 1.3 V returned information about the behaviour of the cell in terms of any possible activation, ohmic and diffusive constraints, the effect of temperature on the OCV and on the kinetics of reactions, etc. Such tests are reported in the Supplementary Materials since this paper is mainly devoted to the discussion of the GC results and to the findings of the most promising operating conditions. This range of temperatures and the current density circulated in the cell under operating conditions were selected according to the characteristics of the cell. Below the lower temperature investigated (i.e., 525 °C), too-high constraints derived from the ohmic resistance of the electrolyte and from the overpotential

due to the evolution of oxygen (i.e., reaction occurring at the anode of cell) did not permit the circulation of an adequate current. At 525 °C, the maximum current density permitted was 150 mA cm$^{-2}$. Above this value, a voltage higher than 2 V was recorded, and this value did not comply with the efficient use of this technology and with the risk of cell and housing degradation. However, this value of current is still acceptable, since it is close to the practical long-term operating condition of commercial cells. Temperatures over 800 °C were not investigated since they are not required by the technological challenges and utilisation, and as a consequence these temperatures, in principle, should be avoided. In addition, the commercial cell selected for this experiment was optimised for operation at an intermediate temperature. In fact, the ASC-400B of Elcogen possess a double thin electrolyte (lower than 5 μm together) and pure cobaltite as an anode, which is currently considered the most performant oxygen electrode. Nevertheless, ASC-400B cells are optimised for SOFC operation and are simply adapted to SOEC operation. However, a relapse of this adaptation consists of an anode–electrolyte interface morphology not fully optimised for a high current density. In the literature, the high risks of delamination occurring at the anode–electrolyte interface, where a high flow of molecular oxygen (occurring at high current density) is literally "bubbling" during its evolution at the anode side, are reported [51,52]. Based on these considerations, a fixed current density of 150 mA cm$^{-2}$ has been adopted for the tests.

Figure 1 depicts the overall operation time for the cell. We observe that both the OCV and cell voltage at 150 mA cm$^{-2}$ were strongly affected by the increased temperature. As observed, the cell operates above the thermoneutral potential only at 525 °C, whereas from 550 °C the cell operates at a voltage below the thermoneutral potential [53]. The decrease in OCV with increased temperatures is a consequence of the reversible potential due to the $H_2$/$O_2$ reaction. Instead, the decrease in cell voltage at 150 mA cm$^{-2}$ is due to the positive effects of increased temperature on both the activation and ohmic constraints, as was also proven by the EIS and I/V tests reported in the SI. Furthermore, the OCV recorded up to 700 °C was close to 1 V as a proof of the limited or absent gas leakages from the cathode chamber. Nevertheless, at higher temperatures, the significantly increased noise and relevant cell voltage loss suggest a chemical degradation of the cell due to the increased kinetics for the re-oxidation of Ni as a consequence of the low amount of $H_2$ and the presence of the $H_2O$ and $CO_2$ fed to the cathode (Equations (1) and (2)). However, the re-oxidation of Ni as a consequence of the possible leakage of $O_2$ from the atmospheric air is negligible because of the high OCV recorded (Figure 1) and because the carbon balance was close to 1 at all temperatures investigated (see the supporting information).

$$Ni + H_2O \rightarrow NiO + H_2 \tag{1}$$

$$Ni + CO_2 \rightarrow NiO + CO \tag{2}$$

In Table 1 we report the main electrochemical results achieved for the bare and coated cells. It is informative as the coated cell showed a significantly lower performance, especially at intermediate temperatures. Such behaviour is a direct consequence of the limited conductivity of perovskite compared to Ni-YSZ, as is proven by the differences in the area-specific resistance (ASR) values reported in Table 1.

The results of gas-chromatographic experiments were treated according to the following equations:

$$CO_2 \; conversion \; (\%) = \frac{CO_{2_{in}} - CO_{2_{out}}}{CO_{2_{in}}} \tag{3}$$

$$H_2 \; residue \; (\%) = \frac{H_{2_{out}}}{H_{2_{in}}} \tag{4}$$

$$Selectivity \; to \; CO \; (\%) = \frac{CO_{out}}{CO_{2_{in}}} \tag{5}$$

$$CO \; yield \; (\%) = CO_2 \; conversion * CO \; selectivity \tag{6}$$

$$Selectivity\ to\ CH_4(\%) = \frac{CH_{4out}}{CO_{2in}} \tag{7}$$

$$CH_4\ yield\ (\%) = CO_2\ conversion * CH_4\ selectivity \tag{8}$$

These results were compared with the theoretical data at the OCV (i.e., without the ionic oxygen moved from the cathode to the anode through the electrolyte) obtained using the GASEQ software.

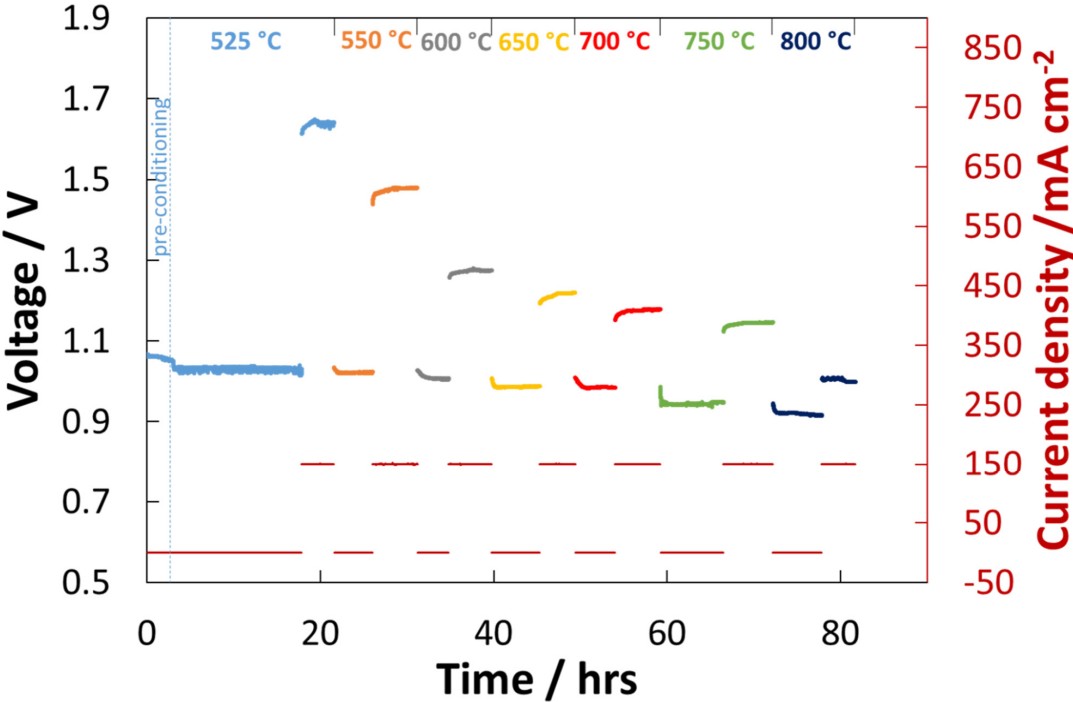

**Figure 1.** Life-time test for the coated cell investigated in the temperature range 525–800 °C for the co-electrolysis of $H_2O$ and $CO_2$.

**Table 1.** Most relevant electrochemical results achieved with bare and coated cells.

| | | Temperature | | | | | | |
|---|---|---|---|---|---|---|---|---|
| | | 525 °C | 550 °C | 600 °C | 650 °C | 700 °C | 750 °C | 800 °C |
| Cell Voltage at 150 mA cm$^{-2}$ | *Coated cell* | 1.88 V | 1.48 V | 1.24 V | 1.24 V | 1.24 V | 1.11 V | 1.00 V |
| | *Bare cell Ref.* [50] | 1.15 V | 1.07 V | 0.99 V | 0.94 V | 0.91 | n.a. | n.a. |
| Area Specific Resistance at 150 mA cm$^{-2}$ | *Coated cell* | >16 Ω cm$^2$ | 3.51 Ω cm$^2$ | 2.07 Ω cm$^2$ | 1.80 Ω cm$^2$ | 1.72 Ω cm$^2$ | 1.33 Ω cm$^2$ | 0.69 Ω cm$^2$ |
| | *Bare cell Ref.* [50] | 2.89 Ω cm$^2$ | 2.25 Ω cm$^2$ | 1.24 Ω cm$^2$ | 0.81 Ω cm$^2$ | 0.76 Ω cm$^2$ | n.a. | n.a. |

The experimental data treated according to Equation (3) and the simulated data for similar conditions are reported in Figure 2. Chemical (Equations (2) and (9)–(11)) and electrochemical reactions (Equations (12)–(14)) were involved in the $CO_2$ conversion. An error bar related to the systematic errors occurring during the GC analysis was added. In this figure, it is worth noting the low reactivity of $CO_2$ and the sensible deviation from the thermodynamic values, especially at lower temperatures. Nevertheless, the circulation of current has increased the conversion of $CO_2$ as a consequence of the simultaneous occurrence of its electrochemical (Equation (12)) and chemical (Equation (9)) reduction. In principle, the high temperatures and the increased partial pressure of $H_2$ produced by the electrochemical reduction of $H_2O$ (Equation (15)) moved the chemical reduction of $CO_2$ to CO (Equation (9)). In addition, a significant increase in this trend was observed at

temperatures above 700 °C due to the increased kinetic involving the reaction between $CO_2$ and Ni to form NiO (Equation (2)).

$$CO_2 + H_2 \; \rightarrow CO + H_2O \tag{9}$$

$$CO_2 + 2H_2 \; \rightarrow C + 2H_2O \tag{10}$$

$$CO_2 + 4H_2 \; \rightarrow CH_4 + 2H_2O \tag{11}$$

$$CO_2 + 2e^- \; \rightarrow CO + O^{2-} \tag{12}$$

$$CO_2 + 4e^- \; \rightarrow C + 2O^{2-} \tag{13}$$

$$CO_2 + 2H_2O + \; 8e^- \; \rightarrow CH_4 + 4O^{2-} \tag{14}$$

These results were similar to those achieved with the bare cell up to 650 °C, and prove that the functional layer was not extremely active at intermediate temperatures (Table 2). A significant increase is instead observed at temperatures above 700 °C as clear evidence of the role played by the functional layer.

**Table 2.** Most relevant chromatographic data achieved with bare and coated cells.

| | | Temperature | | | | | | |
|---|---|---|---|---|---|---|---|---|
| | | **525 °C** | **550 °C** | **600 °C** | **650 °C** | **700 °C** | **750 °C** | **800 °C** |
| **Coated cell -@ 150 mA cm$^{-2}$-** | *$CO_2$ conversion* | 14.1% | 15.4% | 19.1% | 32.5% | 57.9% | 66.8% | 66.3% |
| | *$H_2$ residue* | 89.2% | 89.2% | 88.4% | 85.2% | 80.11% | 71.9% | 67.4% |
| | *Selectivity to CO* | 92.4% | 96.7% | 99.1% | 99.8% | 99.9% | 99.9% | 99.8% |
| | *CO yield* | 13.1% | 14.9% | 19.0% | 32.5% | 57.8% | 66.8% | 66.1% |
| | *Selectivity to $CH_4$* | 7.6% | 3.2% | 0.8% | 0.2% | 0.1% | 0.1% | 0.2% |
| | *$CH_4$ yield* | 1.1% | 0.5% | 0.2% | 0.1% | 0.1% | 0.1% | 0.1% |
| **Bare cell -@ 150 mA cm$^{-2}$- Ref. [50]** | *$CO_2$ conversion* | 25.9% | 26.9% | 29.8% | 30.3% | 31% | n.a. | n.a. |
| | *$H_2$ residue* | 107% | 108.1% | 108.2% | 103.7% | 98.9% | n.a. | n.a. |
| | *Selectivity to CO* | 97.9% | 99.2% | 99.9% | 100% | 100% | n.a. | n.a. |
| | *CO yield* | 25.4% | 26.7% | 29.7% | 30.3% | 30.9% | n.a. | n.a. |
| | *Selectivity to $CH_4$* | 2.1% | 0.8% | 0.1% | 0% | 0% | n.a. | n.a. |
| | *$CH_4$ yield* | 0.5% | 0.2% | traces | traces | traces | n.a. | n.a. |

The evaluation of $H_2$ content in the outlet gas is presented in Figure 3, where the comparison of results achieved under two practical conditions (i.e., OCV and 150 mA cm$^{-2}$) and the data predicted thermodynamically are reported. Under the circulation of current, the reduction in $H_2O$ is expected:

$$H_2O + \; 2e^- \; \rightarrow \; H_2 + O^{2-} \tag{15}$$

This figure shows two important aspects of the experimental results. The first concerns the deviations at low temperatures between the bar charts under practical conditions and thermodynamic values that are mainly due to the limited $CO_2$ chemical reduction as observed in Figure 2. The second aspect is related to the $H_2$ residue that is less than 100% under practical conditions (compared to the inlet), and this means that $H_2$ is consumed

by the reduction of $CO_2$ and CO, and by the maintaining of Ni in the metallic state. These reactions are temperature-activated and justify the trends observed for the conversion of $CO_2$ (Figure 1). Additionally, the contrary trends observed for the thermodynamic and practical conditions bar charts are evidence of $H_2$ demand due to the parasitic reactions occurring under practical conditions and involving the Ni. Here, the behaviour of this cell was similar to that of the bare cell, demonstrating the limited promoting role of the functional layer towards the reduction of $H_2O$ (Table 2).

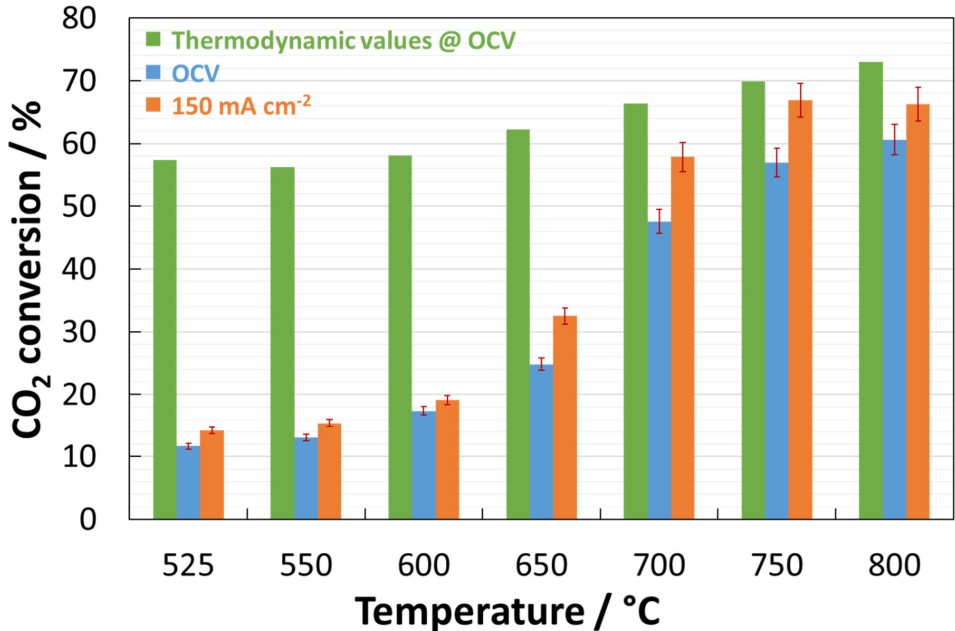

**Figure 2.** $CO_2$ conversion based on thermodynamic prediction and under practical conditions treated according to Equation (3).

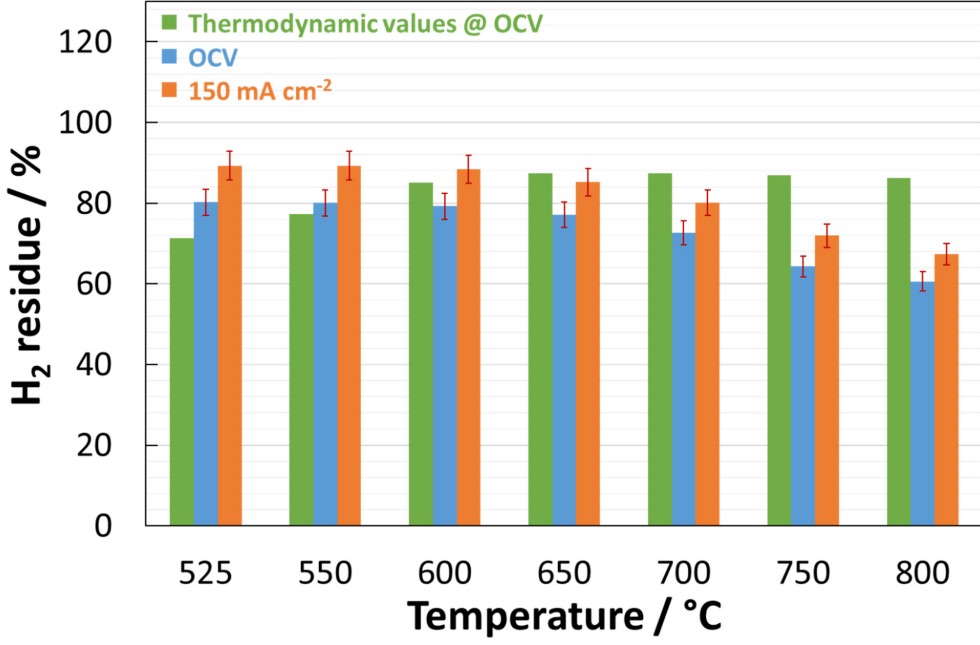

**Figure 3.** Trend of $H_2$ residue based on thermodynamic prediction and under practical conditions treated according to Equation (4).

The data related to CO residue were treated according to Equations (5) and (6). Figure 4a depicts the trend for selectivity to CO in the temperature range of 525–800 °C. As shown, the cell promoted a high selectivity to CO, as was expected from its favourable thermodynamic formation with increasing temperatures. At 525 °C, the selectivity to CO was approximately 95% under OCV and 92% under the circulation of current, which was higher than expected on the basis of the thermodynamic results. By increasing the temperature, these values increased too and reached 100% at 650 °C. However, one positive aspect was that the circulation of the current slightly depleted the selectivity to CO. Such behaviour was evident at 525 °C, and indicates the minor contribution of Equation (12) to the overall conversion of $CO_2$.

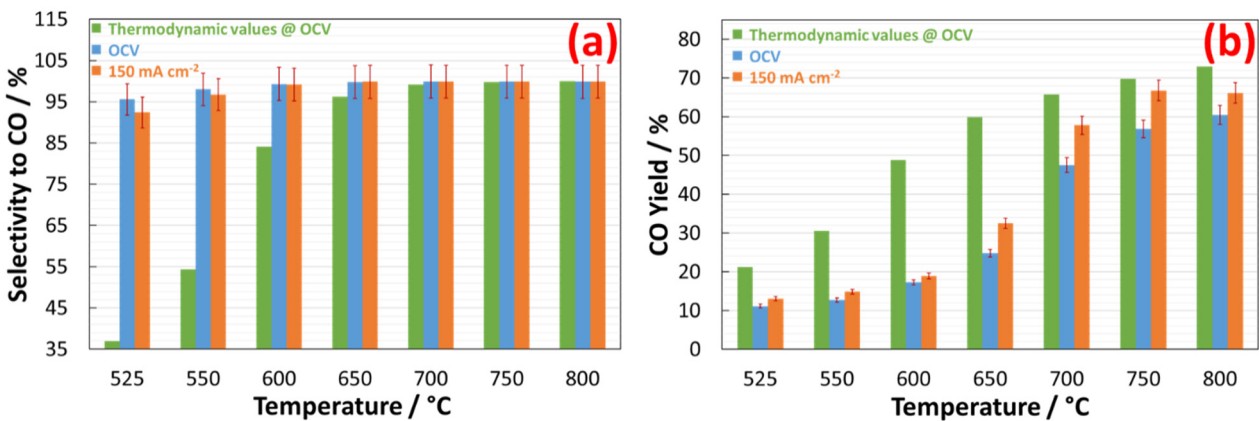

**Figure 4.** Trend of CO selectivity (**a**) and yield (**b**) treated according to Equations (5) and (6).

The Figure 4b reports the CO yield evaluated according to Equation (6). The trends of bar charts for the experimental data were in agreement with the thermodynamic data. Nevertheless, the yield of CO observed in the outlet gas was significantly lower than the predicted yield as a consequence of the low reactivity of $CO_2$, as discussed in Figure 2. Furthermore, the positive effect on the CO yield observed for the study case with a current density of 150 mA cm$^{-2}$ was a consequence of the increased $H_2$ partial pressure in reaction 9, produced from the direct conversion of $H_2O$ (Equation (15)).

Concerning the $CH_4$ residue in the outlet gas, the data were treated according to Equations (7) and (8). Figure 5a shows that the selectivity to $CH_4$ is relatively low and is limited to the intermediate temperatures. This behaviour is in agreement with the thermodynamic trend for methane stability, which requires temperatures below 650 °C in order to become favourable compared to CO. Nevertheless, the circulation of the current in the cell at 525 °C promoted a significant increase in selectivity to methane. Although this behaviour can be ascribed to the direct methanation of $CO_2$ (Equation (14)), the low electrochemical reactivity of $CO_2$ (see Figure 2) suggests that the $CH_4$ was produced through the combination of $CO_2$ and $H_2$. As discussed for the production of CO (Figure 3), a consequence of the increased $H_2$ partial pressure promoted by the electrochemical reduction of $H_2O$ (Equation (19)) can affect the quality of the outlet gas. However, the $CH_4$ yield (Figure 5b) measured during the cell operation was significantly lower than the predicted values, and this is the direct consequence of the low reactivity of $CO_2$ to its catalytic and electrocatalytic reduction.

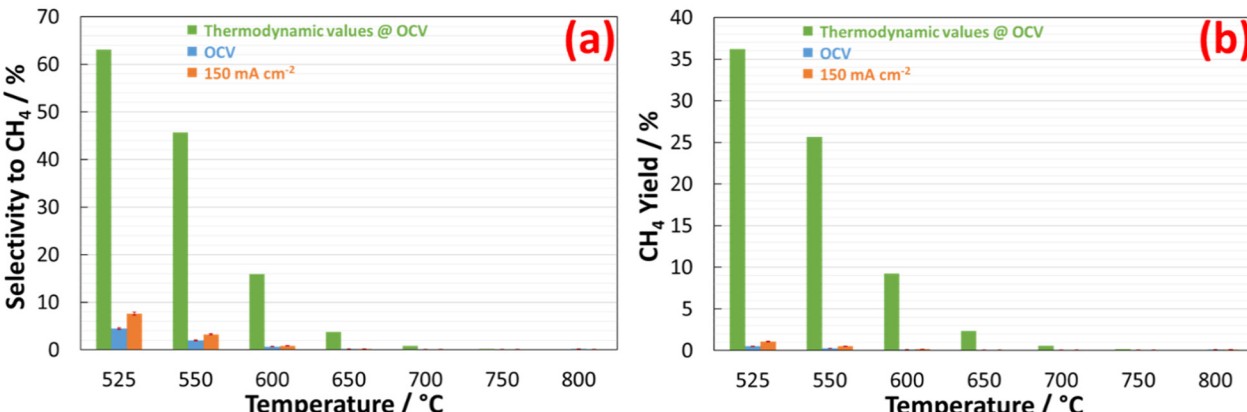

**Figure 5.** Trend of $CH_4$ selectivity (**a**) and yield (**b**) treated according to Equations (7) and (8).

In terms of trends, the bar charts reported for the CO (Figure 4) and $CH_4$ (Figure 5) were similar to that achieved for a bare cell. Nevertheless, by comparing the absolute values recorded in these two experiments, it was proven that the addition of Ni-modified LSFCO increased the quality of the outlet gas about twofold in terms of $CH_4$ yield (Table 2).

In Table 2 are summarised the results of the gas-chromatographic tests concerning bare and coated cells under a current density of 150 mA cm$^{-2}$. As discussed, the addition of a coating layer to the bare cell was not particularly effective for a significant increase in $CO_2$ and $H_2O$ conversion at temperatures below 600 °C, although at higher temperatures the promoting effect of perovskite was observed. Nevertheless, this is sensible as the coating layer promoted an increased quality of gas as a consequence of the reduced amount of CO and the increased amount of $CH_4$.

Concerning the possible formation of carbon as a consequence of the chemical (Equation (10)) and electrochemical (Equation (13)) reactions, the sum of C species determined in the gas effluent was closed to 100% (see the SI).

An indication of the efficiency of the chemical and electrochemical reactions related to the production of CO and $CH_4$ is given through the treatment of the gas-chromatographic data according to the following equations:

$$\xi_{CO_{chem}}(a.u) = \frac{total\ CO\ produced}{H_2\ inlet} \tag{16}$$

$$\xi_{CH_{4chem}}(a.u) = \frac{total\ CH_4\ produced}{4H_2\ inlet} \tag{17}$$

$$\xi_{CO_{Faradaic}}(a.u.) = \frac{total\ CO\ produced}{CO\ electrochem\ obtainable} \tag{18}$$

$$\xi_{CH_{4Faradaic}}(a.u.) = \frac{total\ CH_4\ produced}{CH_4\ electrochem\ obtainable} \tag{19}$$

where $CO_{electrochem\ obtainable}$ and $CH_{4electrochem\ obtainable}$ are derived according to the following equations:

$$CO_{electrochem\ obtainable}\left(\frac{cc}{min\ cm^2}\right) = \frac{current\ density}{n \cdot F} \tag{20}$$

$$CH_{4electrochem\ obtainable}\left(\frac{cc}{min\ cm^2}\right) = \frac{current\ density}{n \cdot F} \tag{21}$$

The current density circulated into the cell was 150 mA cm$^{-2}$. The term "n" is the number of electrons involved in the reducing reaction of $CO_2$. In the case of CO, n is equal to 2, according to Equation (16), whereas in the case of $CH_4$, n is equal to 8 according to Equation (18). F is the Faradaic constant.

The $\zeta_{X_{chem}}$ provides an indication of how effective the chemical reaction is in producing the $X$ species, and $\zeta_{X_{Faradaic}}$ is estimated on the basis of the theoretical electrochemical production of $X$ species. Values $\zeta > 1$ are possible only if the electrochemical reduction of $CO_2$ adds to the CO or $CH_4$ formation. In the contrary case, the electrochemical conversion of $CO_2$ is less probable, but cannot be excluded.

The bar charts reported in Figure 6a represent the efficiency of CO formation (Equations (16) and (18)) expressed in arbitrary units (a.u.). As shown, the data at OCV are close to the thermodynamic values, but with the circulation of current in the cell, a great enhancement in the efficiency of CO yield was observed. At 150 mA cm$^{-2}$, the electrochemical reduction of both $CO_2$ (Equation (12)) and $H_2O$ (Equation (15)) is expected, although, as discussed above, the first reaction is limited, especially at lower temperatures. As a consequence, the bar charts at 150 mA cm$^{-2}$ at lower temperature are enhanced due to the reverse water–gas shift reaction (Equation (9)), whereas, starting from 650 °C, the direct electrochemical conversion of $CO_2$ to CO becomes dominant (Equation (12)), since $\zeta$ was higher than 1. A similar treatment of the data was developed for the evaluation of the efficiency of methane production according to Equations (17) and (19). As depicted in Figure 6b, the methane produced by a pure chemical reaction is negligible, as the thermodynamics also predict. $CH_4$ is in fact largely unstable at high temperatures, and many parallel reactions (including cracking) concur with its low presence at the outlet. Under the circulation of the current, the concurrent multiple electrochemical reductions of $CO_2$ to CO (Equation (10)) and to C (Equation (11)), along with the high partial pressure of the $H_2$ produced electrochemically, contribute to the formation of methane with an efficiency higher than was predicted.

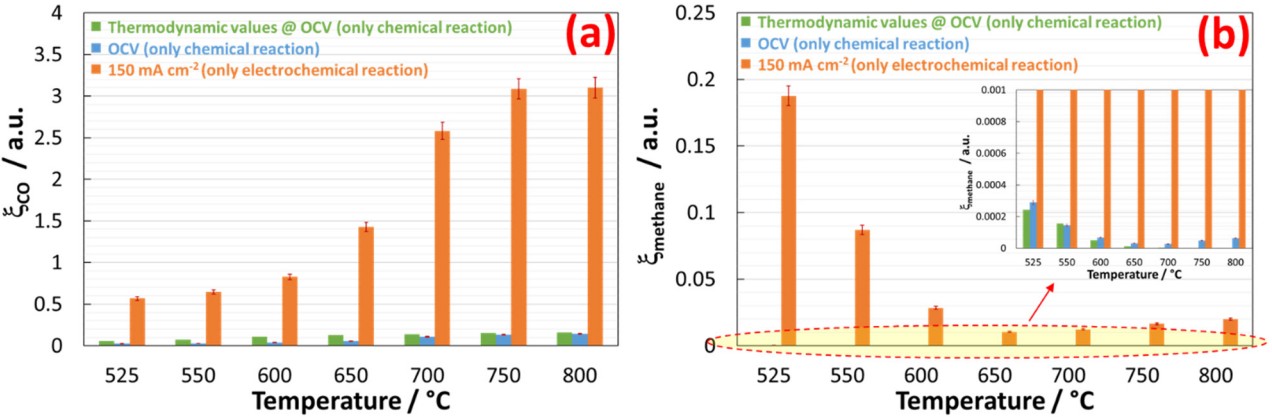

**Figure 6.** Trend of efficiency towards the production of CO (**a**) and methane (**b**) determined using Equations (16)–(19).

## 3. Experimental

The exsolved perovskite used as a functional layer is referred to as Ni-modified $La_{0.6}Sr_{0.4}Fe_{0.8}Co_{0.2}O_3$ (LSFCO). The procedure used refers to the wet impregnation of LSFCO (Praxair, Danbury, CT, USA) with 3 wt. % of Ni (Ni as nitrate, Sigma Aldrich, St. Louis, MO, USA). After drying, the impregnated powders were calcined for 2 h at 500 °C in air, and then reduced for 2 h at 800 °C with diluted $H_2$ (5 vol.%). As a result of thermal treatments, a partial segregation of Fe and Co from the bulk of the perovskite moved to the surface and combined with the Ni to form fine embedded nanoparticles on the surface of the depleted LSFCO. The resulting phases were a Ruddlesden–Popper-type structure [41,54,55] and a solid oxide solution between Ni, Fe, and Co (i.e., $\alpha$-$Fe_{100-y-z}Co_yNi_zO_x$ oxide) [56,57]. This exotic material was extensively investigated by the authors of this work, and is reported in other papers. The functional layer used for this research consisted of a ball-milled mixture of 70 wt. % of Ni-modified perovskite and 30 wt. % of CGO (Sigma Aldrich, St. Louis, MO, USA). Then, the cathode of the SOEC cell was brush-coated with

a slurry of the functional layer, including 3 wt. % of Butvar (Sigma Aldrich, St. Louis, MO, USA) and 1 wt. % $\alpha$-terpineol (Sigma Aldrich, St. Louis, MO, USA), as well as ethanol as solvent in an amount of at least 60 wt. %. The deposit of the functional layer on the cell was 10 mg cm$^{-2}$. The cell used for this experiment was a button cell with an active area of 2 cm$^2$ cut from a large planar cell manufactured by Elcogen-Estonia. This cell is referred to as ASC-400B and consists of a Ni-YSZ cathode, YSZ and CGO double electrolytes and a $La_{0.6}Sr_{0.4}CoO_3$ (LSC) anode. A pre-conditioning step of 2 h carried out at 800 °C in the presence of diluted gas was carried out to promote the exsolution of perovskite and its adhesion to the cathode of the cell. Then, the cell was cooled to 525 °C and the gas inlet was set to $H_2$ (2.5 cc min$^{-1}$ cm$^{-2}$), He (15 cc min$^{-1}$ cm$^{-2}$), $H_2O$ (fed by a syringe pump at 0.005 g h$^{-1}$ cm$^{-2}$ and then vaporised), and $CO_2$ (1 cc min$^{-1}$ cm$^{-2}$) at the cathode, and air (50 cc min$^{-1}$ cm$^{-2}$) at the anode. The cell was experimented upon on a test bench purchased from the Greenlight Innovation-Canada company (Eastlake Campus, BC, Canada). The electrochemical characterisation of the cell was carried out with a BioLogic-France instrument (Seyssinet-Pariset, France), and the dried effluent gas from the cathode was analysed by a micro-(VARIAN-Agilent Technologies, Palo Alto, CA, USA) equipped with Molsieve (20 mt), PoraPLOT Q (10 mt), CB-Sil (8 mt) and micro-DMD (differential mobility detector). The GC results were treated according to a prior calibration carried out with gas calibration mixes purchased from Sigma Aldrich (St. Louis, MO, USA).

## 4. Conclusions

Ni-modified $La_{0.6}Sr_{0.4}Fe_{0.8}Co_{0.2}O_3$ was deeply investigated as an anode for SOFC, and can also be suggested as a cathode for SOECs because of the redox behaviour of perovskite and because commercial cells based on Ni-based cathodes have a high risk of cracking as a consequence of the extensive redox cycle when $H_2O$ is fed into the cell [58]. In this work, we report on the study of a commercial SOEC coated on the cathode, with a functional layer based on this material that was reported in the literature to be exsolved under reducing conditions and effective for the tuning of the outlet gas' quality during the co-electrolysis of $CO_2$ and $H_2O$ [59]. The experiments conducted in the range of temperatures between 525 and 800 °C showed the behaviour of the cell in terms of electrochemical characteristics and gas stream quality from the cathode chamber. The electrochemical behaviour of the cell showed that the addition of a functional layer negatively affected the voltage of the cell as a consequence of the limited conductivity of perovskite. Furthermore, we discussed the role of Ni-modified perovskite in the quality of gas produced during the electrochemical conversion of $CO_2$ and $H_2O$. The treated data concerning the outlet gas stream were discussed in relation to our recent achievements concerning a bare cell investigated under the same conditions. As discussed in this paper, the use of a Ni-modified perovskite reduced the reactivity of $CO_2$, especially at temperatures below 650 °C, where the preferred reaction was the electroreduction of $H_2O$. However, the most significant achievement consisted of the increased gas quality in terms of the increased yield of methane and the depletion of CO at intermediate temperatures, both of which by about twofold, compared to the bare cell. Therefore, the results proved that an optimised exsolved perovskite used as a coating layer for an SOEC is an effective approach to the improvement of SOEC technology towards the co-electrolysis of $H_2O$ and $CO_2$.

**Supplementary Materials:** The following are available online at https://www.mdpi.com/2073-4344/11/1/56/s1, Figure S1. Polarization curves of the coated cell investigated in the temperature range 525–800 °C for the co-electrolysis of $H_2O$ and $CO_2$. Figure S2. Impedance spectra of the coated cell investigated at OCV in the temperature range 525–800 °C for the co-electrolysis of $H_2O$ and $CO_2$. Figure S3. Impedance spectra of the coated cell investigated at 1.3 V in the temperature range 525–800 °C for the co-electrolysis of $H_2O$ and $CO_2$. Figure S4. Analytes separated with a Molsieve Column and revealed with TCD. Gas-Chromatograms of outlet from cathode chamber of cell operating at 525 °C operating at 150 mA cm$^{-2}$ (a) and under OCV (b). Figure S5. Analytes separated with a Pore-Plot Q Column and revealed with TCD. Gas-Chromatograms of outlet from cathode chamber of cell operating at 525 °C operating at 150 mA cm$^{-2}$ (a) and under OCV (b).

Figure S6. Carbon balance of gas achieved under practical conditions. The gas-chromatographic data were treated ac-cordingly to Equation (1). Figure S7. SEM analysis of spent cell highlighting the interface regions (a) and the magnification of cathode microstruc-ture (b). Figure S8. XRD spectrum of functional layer on the spent cell. Figure S9. EDX analysis of spent functional layer.

**Author Contributions:** Conceptualisation, A.S.A. and M.L.F.; methodology, A.S.A. and S.C.Z.; software, M.L.F.; analysis, S.C.Z. and M.L.F.; investigation, S.C.Z.; writing—original draft preparation, M.L.F. and A.S.A.; writing—review and editing, M.L.F. and A.S.A.; project administration, A.S.A.; funding acquisition, V.A. All authors have read and agreed to the published version of the manuscript.

**Funding:** The present work was carried out within an Agreement between the Italian Ministry of Economic Development (MISE) and the National Research Council (CNR) in the framework of a Research Program for the Electric System (RdS-PAR2019).

**Institutional Review Board Statement:** Not applicable.

**Informed Consent Statement:** Not applicable.

**Data Availability Statement:** The data presented in this study are available in the article.

**Conflicts of Interest:** The authors declare no conflict of interest.

## Abbreviations

| Acronym | Full Form |
| --- | --- |
| ASC | Anode-supporting cell |
| ASR | Area-specific resistance |
| a.u. | arbitrary unit |
| CGO | $Ce_{0.9}Gd_{0.1}O_{2-\delta}$ |
| GC | Gas-chromatography |
| GDC | Gadolinia-doped ceria |
| LSC | $La_{0.6}Sr_{0.4}CoO_3$ |
| LSFCO | $La_{0.6}Sr_{0.4}Fe_{0.8}Co_{0.2}O_3$ |
| MIEC | Mixed ionic and electronic conductor |
| n.a. | data not available |
| OCV | Open circuit voltage |
| SOEC | Solid oxide electrolysis cell |
| SOFC | Solid oxide fuel cell |
| SOC | Solid oxide cell |
| YSZ | Yttria-stabilised zirconia |

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
