# Peer review of "The Effect of Ni-Modified LSFCO Promoting Layer on the Gas Produced through Co-Electrolysis of CO2 and H2O at Intermediate Temperatures"

_catalysts, doi:10.3390/catal11010056_

Round 1

Reviewer 1 Report

The present article deals with the co-electrolysis of CO2/H2O in a solid oxide electrolyzer, as a function of temperature (in the range 525-800°C] onto a Ni-modified LSFCO electrode. The outlet gas analysis has been carried out by gas chromatography (GC), both at OCP and under polarization conditions.

If this hot topic is of interest, especially the electrochemical cell-GC coupling to analyze the out gas, the paper should contain more details results. I recommend major correction before publication.

General comments:

Abbreviations list is useful, nevertheless, the authors use both CGO and GDC for gadolina doped ceria and only GDC is defined and listed.

Regarding equations: it should be useful to define each terms of equation, with units (e.g. Eq 5, 6…Eq 20, 21 …)

For Eq 24 and 25, the authors should right the general equation (not with the values directly written) and then given the definition and units of general terms.  Moreover, the molar volume (24.4 L/mol) is used but this value change with temperature… The authors should discuss that and justify the chosen temperature.

The introduction presents more than 1 page about the previous results of authors themselves. This could be reduced and summarized (especially the ethanol part). Note that 1/3 of references are self-citations… Authors are very efficient and present in that field, nevertheless this ratio may be too high.

Even if the paper aims at GC analysis, a polarization curves would be useful, at least to compare with a bare electrode. (even in a table form)

Regarding GC analysis, no chromatogram is given, no retention time, no calibration curves, etc. As the paper focuses on GC, at least on graph should be presented and discussed.

The GC results are discussed regarding different chemical reaction. In the temperature range [525-800°C], the Gibbs energy of the reaction change… from thermodynamically favorable to unfavorable and vice versa. It could be more relevant to discuss the variation of gas conversion and selectivity with regards to thermodynamics data.

The influence of Ni-coated cathode has to be compared to previous results obtained on bare cathode. A table could help to sum up the results and enhance the differences/improvement.

In conclusion, there is a part discussion the electrochemical characteristics of coated/uncoated cathode but none is presented in that paper… neither the delamination process… please add results or write the conclusion in accordance with the presented results.

For all of that comments, I recommend major modifications before publication.

Author Response

Reviewer #1: 

The present article deals with the co-electrolysis of CO2/H2O in a solid oxide electrolyzer, as a function of temperature (in the range 525-800°C] onto a Ni-modified LSFCO electrode. The outlet gas analysis has been carried out by gas chromatography (GC), both at OCP and under polarization conditions.

If this hot topic is of interest, especially the electrochemical cell-GC coupling to analyze the out gas, the paper should contain more details results. I recommend major correction before publication.

General comments:

Abbreviations list is useful, nevertheless, the authors use both CGO and GDC for gadolina doped ceria and only GDC is defined and listed.

We thank the reviewer for this suggestion. The revised text now includes CGO in the “Abbreviations” (Page 18)

Regarding equations: it should be useful to define each terms of equation, with units (e.g. Eq 5, 6…Eq 20, 21 …)

The equations related to the treatment of gas-chromatographic data were renumbered (3-8 in the revised text). These now include “%” (page 7).

Concerning the eqs. 20-23, these were renumbered to 16-19 and reported in arbitrary units (a.u.). (pages 13-14).

For Eq 24 and 25, the authors should right the general equation (not with the values directly written) and then given the definition and units of general terms.  Moreover, the molar volume (24.4 L/mol) is used but this value change with temperature… The authors should discuss that and justify the chosen temperature.

Thank you so much for this observation that gave us to rectify a gross mistake in the formula. The text was improved accordingly. (page 14)

The introduction presents more than 1 page about the previous results of authors themselves. This could be reduced and summarized (especially the ethanol part). Note that 1/3 of references are self-citations… Authors are very efficient and present in that field, nevertheless this ratio may be too high.

The revised manuscript was cleared of this part.

Even if the paper aims at GC analysis, a polarization curves would be useful, at least to compare with a bare electrode. (even in a table form)

Thank you for this suggestion. The revised text includes a table collecting the main electrochemical behaviour of bare and coated cells. (page 7)

Regarding GC analysis, no chromatogram is given, no retention time, no calibration curves, etc. As the paper focuses on GC, at least on graph should be presented and discussed.

The revised text reports about the column used for these analyses (page 16) and the supplementary materials now include the gas chromatograms carried out at 525 °C.

The GC results are discussed regarding different chemical reaction. In the temperature range [525-800°C], the Gibbs energy of the reaction change… from thermodynamically favorable to unfavorable and vice versa. It could be more relevant to discuss the variation of gas conversion and selectivity with regards to thermodynamics data.

Thank you for this observation. The figures 4 and 5 now include the thermodynamic values and specific comments were included in the revised manuscript.

The influence of Ni-coated cathode has to be compared to previous results obtained on bare cathode. A table could help to sum up the results and enhance the differences/improvement.

A table comparing the results achieved with bare and coated cell was included and discussed. (page 13)

In conclusion, there is a part discussion the electrochemical characteristics of coated/uncoated cathode but none is presented in that paper… neither the delamination process… please add results or write the conclusion in accordance with the presented results.

We have revised the conclusions accordingly (pages 16-17)

For all of that comments, I recommend major modifications before publication.

Reviewer 2 Report

The present manuscript claims the effect of Ni-modified LSCFO functional layer on the fuel electrode of SOEC aimed at co-electrolysis application. This is a very well written manuscript with meaningful results. The authors did excellent job in explaining the effect of modified perovskite for the electrochemical conversion of CO2 and H2O including selectivity and efficiency. I accept this manuscript with no revision.

Author Response

Thank you so much for your comments. We are delighted about your opinion on this manuscript

Reviewer 3 Report

I consider an important topic for SOEC development.
However, the paper needs major revisions.
1-The experimental procedure for the layer adhesion by brush and the sealing method of the SOEC sample is missed in the paper. The layer is calcined in air after brush painting? The electrode is exsolved after adhesion of the layer?
2-Regarding the operation of the cell, a constant current of 150 mA·cm-2 has been applied for all temperatures, however, below 650 ºC the system is working above the thermoneutral voltage. Thus, the system is not working in the correct operation conditions. Furthermore, the Faraday’s efficiency is missed and should be introduced ( to make clear the electrolysis process and efficiency, in order to check possible electronic leakages).
3-The purpose of the paper is not clear for me. It is said that the co-electrolysis of CO2 and H2O is improved by the incorporation of this layer, but if we check the H2residue, H2 is not produced and H2 (feed in the system) is consumed in the system (CO2 reduction). Then, higher amount of CO is obtained, as a result the syngas ratios are not achieved. Furthermore, the CO2 or CO hydrogenation takes place at temperatures below 500ºC, achieving good results in terms of selectivity at temperatures around 350 ºC with high ratios of H2:CO2. This criteria is thermodynamically fixed, for that reason, the selectivity is as low as presented.
4-Details like SEM images, XRD or the Impedance results are missed in the paper, they should be included to give more information about the influence of this layer on the gas analysis results.
5- Finally, along all paper, the sample presented here is compared with the bare sample, but results (partial or total) are not presented in the paper. Difficult to observe the advantages of the layer incorporation.

Author Response

Reviewer #3: 

I consider an important topic for SOEC development. However, the paper needs major revisions.

1-The experimental procedure for the layer adhesion by brush and the sealing method of the SOEC sample is missed in the paper. The layer is calcined in air after brush painting? The electrode is exsolved after adhesion of the layer?

Thank you for this comment. The text was improved accordingly (page 16).

2-Regarding the operation of the cell, a constant current of 150 mA·cm-2 has been applied for all temperatures, however, below 650 ºC the system is working above the thermoneutral voltage. Thus, the system is not working in the correct operation conditions. Furthermore, the Faraday’s efficiency is missed and should be introduced ( to make clear the electrolysis process and efficiency, in order to check possible electronic leakages).

Thank you for these comments.

At 650 °C and 150 mA cm-2, the cell operates at about 1.22 V that is below the thermoneutral potential of about 1.5 V in this temperature range, as reported in the following paper: Progress in Energy and Combustion Science, Volume 36, Issue 3, June 2010, Pages 307-326.

In principle, in the experiments reported in this manuscript, only at 525 °C the cell operates over the thermoneutral pontential.

Then, in general, all low temperature electrolysis systems (e.g. PEM and alkalyne based systems) operate above the tehrmoneutral potential (i.e. 1.47-1.5 V).

Concerning the SOECs, generally such technology operates below the thermoneutral potential because they operate at low current density and the reversible heat (TΔS°) is supplied externally by the furnace.

Please see the “EU harmonised terminology for low-temperature water electrolysis for energy-storage applications” where heat imput is considered to explain how this can occur.

Concerning the Faradaic efficiency, it is unfortunately affected by the parasitic reactions involving the cell and in particular to the reoxidation–reduction occurring to the Ni of cathode.

Therefore, no accurate estimation is possible in this case.

Specific sentences were added to the revised text.

3-The purpose of the paper is not clear for me. It is said that the co-electrolysis of CO2 and H2O is improved by the incorporation of this layer, but if we check the H2residue, H2 is not produced and H2 (feed in the system) is consumed in the system (CO2 reduction). Then, higher amount of CO is obtained, as a result the syngas ratios are not achieved. Furthermore, the CO2 or CO hydrogenation takes place at temperatures below 500ºC, achieving good results in terms of selectivity at temperatures around 350 ºC with high ratios of H2:CO2. This criteria is thermodynamically fixed, for that reason, the selectivity is as low as presented.

Some of H2 fed into the system or produced is used to reduce the NiO to metallic Ni and this affects the results. Nevertheless, we agree with the reviewer that some improvements are necessary. This behaviour of cell was discussed in the text and further specific sentences were added to the revised text.

4-Details like SEM images, XRD or the Impedance results are missed in the paper, they should be included to give more information about the influence of this layer on the gas analysis results.

5- Finally, along all paper, the sample presented here is compared with the bare sample, but results (partial or total) are not presented in the paper. Difficult to observe the advantages of the layer incorporation.

Please accept our apologies if we commented these two points together.

These experiments were reported in the supplementary information.

Also, to comply with the suggestions raised from reviewer 1, we have added two tables in the revised manuscript concerning the electrochemical experiments

Round 2

Reviewer 3 Report

I really appreciate your comments and revision, but I am missed more information regarding de final composition of the coated layer. In the paper is said that this material presents exsolution, but not images are presented, none XRD are showed of the coated layer after test.

Author Response

I really appreciate your comments and revision, but I am missed more information regarding de final composition of the coated layer. In the paper is said that this material presents exsolution, but not images are presented, none XRD are showed of the coated layer after test.

We thank the reviewer for this useful comment. The revised text now includes the content of deposited functional layer (page 16), whereas the supporting materials include the XRD and TEM of spent functional layer.